# Coumarin-Based Sulfonamide Derivatives as Potential DPP-IV Inhibitors: Pre-ADME Analysis, Toxicity Profile, Computational Analysis, and In Vitro Enzyme Assay

**DOI:** 10.3390/molecules28031004

**Published:** 2023-01-19

**Authors:** Pallavi Kishor Vawhal, Shailaja B. Jadhav, Sumit Kaushik, Kahnu Charan Panigrahi, Chandan Nayak, Humaira Urmee, Sharuk L. Khan, Falak A. Siddiqui, Fahadul Islam, Aziz Eftekhari, Abdullah R. Alzahrani, Mohd Fahami Nur Azlina, Md. Moklesur Rahman Sarker, Ibrahim Abdel Aziz Ibrahim

**Affiliations:** 1Department of Pharmaceutical Chemistry, PEA’s Modern College of Pharmacy, Sector 21, Yamunanagar, Nigdi 411044, India; 2Faculty of Pharmacy, Raja Balwant Singh Engineering Technical Campus, Bichpuri, Agra 283105, India; 3Department of Pharmaceutics, Gayatri Institute of Science and Technology, Gyan Vihar, Gunupur 765022, India; 4Department of Pharmaceutics, School of Pharmaceutical Education and Research, Berhampur University, Ganjam 760007, India; 5Department of Pharmaceutical Science, North South University, Dhaka 1229, Bangladesh; 6Department of Pharmaceutical Chemistry, N.B.S. Institute of Pharmacy, Ausa 413520, India; 7Department of Pharmacy, Faculty of Allied Health Sciences, Daffodil International University, Dhaka 1207, Bangladesh; 8Research Center for Pharmaceutical Nanotechnology, Biomedicine Institute, Tabriz University of Medical Sciences, Tabriz 51665118, Iran; 9Department of Pharmacology and Toxicology, Faculty of Medicine, Umm Al-Qura University, Makkah 4089-6928, Saudi Arabia; 10Department of Pharmacology, Faculty of Medicine, University Kebangsaan Malaysia, Jalan Yacob Latif, Kuala Lumpur 56000, Malaysia; 11Department of Pharmacy, State University of Bangladesh, 77 Satmasjid Road, Dhanmondi, Dhaka 1205, Bangladesh; 12Health Med Science Research Network, 3/1, Block F, Lalmatia, Dhaka 1207, Bangladesh

**Keywords:** ADMET, molecular docking, sitagliptin, hypoglycemia, SAR, DPP-IV, coumarin-based sulfonamide, computation analysis, docking, in vitro, toxicity

## Abstract

Recent research on dipeptidyl peptidase-IV (DPP-IV) inhibitors has made it feasible to treat type 2 diabetes mellitus (T2DM) with minimal side effects. Therefore, in the present investigation, we aimed to discover and develop some coumarin-based sulphonamides as potential DPP-IV inhibitors in light of the fact that molecular hybridization of many bioactive pharmacophores frequently results in synergistic activity. Each of the proposed derivatives was subjected to an in silico virtual screening, and those that met all of the criteria and had a higher binding affinity with the DPP-IV enzyme were then subjected to wet lab synthesis, followed by an in vitro biological evaluation. The results of the pre-ADME and pre-tox predictions indicated that compounds 6e, 6f, 6h, and 6m to 6q were inferior and violated the most drug-like criteria. It was observed that 6a, 6b, 6c, 6d, 6i, 6j, 6r, 6s, and 6t displayed less binding free energy (PDB ID: 5Y7H) than the reference inhibitor and demonstrated drug-likeness properties, hence being selected for wet lab synthesis and the structures being confirmed by spectral analysis. In the in vitro enzyme assay, the standard drug Sitagliptin had an IC_50_ of 0.018 µM in the experiment which is the most potent. All the tested compounds also displayed significant inhibition of the DPP-IV enzyme, but 6i and 6j demonstrated 10.98 and 10.14 µM IC_50_ values, respectively, i.e., the most potent among the synthesized compounds. Based on our findings, we concluded that coumarin-based sulphonamide derivatives have significant DPP-IV binding ability and exhibit optimal enzyme inhibition in an in vitro enzyme assay.

## 1. Introduction

In type 2 diabetes mellitus (T2DM), either the body does not produce sufficient insulin or it becomes resistant to insulin. The current oral care therapeutics for T2DM attempt to reduce the amount of glucose that is synthesized in the liver, raise the amount of glucose that is secreted, decrease the amount of glucose that is consumed, and maximize the use of glucose that is left over [1,2,3]. T2DM is treated with sulfonylureas, thiazolidinediones, biguanides, alpha-glucosidase inhibitors, and other drugs that have been linked to a wide range of side effects, such as low blood sugar and weight gain [4,5,6].

Several new anti-diabetic medications, such as 11b-hydroxysteroid dehydrogenase 1 blockers, sodium-glucose co-transporter 2 blockers, insulin analogues, glucagon antagonists, and dipeptidyl peptidase-IV (DPP-IV) inhibitors, have been discovered to address these undesirable effects [7]. Glucokinase activators and insulin-releasing enhancers are two examples of these novel therapies [8,9]. As a consequence of this, DPP-IV inhibitors have been found to be risk-free and successful in the treatment of T2DM, leading to their widespread usage [1,10,11]. In addition to increasing insulin production, improving β-cell proliferation, and reducing cell death, GLP-1 (glucagon-like peptide-1) is a hypoglycemic incretin that also reduces glucagon and gluconeogenesis when the DPP-IV enzyme is controlled [12,13,14,15,16,17,18].

Until October 2006, sitagliptin (Januvia, Merck; Figure 1) was the only FDA-approved DPP-IV inhibitor medication in the United States [16,18]. It has been shown that they are effective in treating T2DM without causing weight gain or harmful hypoglycemia symptoms [19,20,21,22,23,24]. Research is ongoing to find DPP-IV antagonists that may be used once a week to improve the effectiveness of T2DM treatment [3,25,26,27,28,29,30,31,32]. There were two such medications (Figure 1), Trelagliptin [33] and Omarigliptin [34], available in Japan between March and September of 2015, that obtained recent regulatory clearance.

Coumarins are a type of oxygen heterocyclic molecule that is extremely common. In the development of several high-activity medications, the coumarins’ varied biological actions are taken into consideration [35]. Coumarin derivatives have a broad variety of biological actions, including anticoagulant characteristics and cytotoxic effects on bacteria and other pathogens [36]. These include anti-inflammatory properties, neuroprotection, antioxidant, and anti-hyperglycemic properties, as well as anti-adipogenic and neuroprotective properties [36]. The natural coumarins are stereo-specific and serve as the basis for different alkaloids, macrolides, terpenoids, and pheromones. Designing fluorescent chemosensors, labeling polymers, solar cells, and cell imaging tools are just some of the many applications for coumarin groups [37,38,39,40].

Recently, many researchers in the field have reported that coumarins as potential DPP-IV inhibitors. Rina Soni and Shubhangi S. Soman synthesized and evaluated some 3-aminocoumarin and 7-amino-4-methylcoumarin derivatives for DPP-IV inhibition. At a concentration of 100 µM, three of these compounds exhibited moderate DPP-IV inhibition [41]. In another study, Radhika Sharma and Shubhangi S. Soman synthesized 3-aminocoumarin derivatives as potential DPP-IV inhibitors and reported one potent compound with a 3.16 ± 1.28 µM of IC_50_ value [42]. Rina Soni et al. designed and synthesized some chromen-2-one derivatives and evaluated them as potential DPP-IV inhibitors; out of these two compounds, quite good inhibition was observed at 10 μM concentration [43]. Anand-Krishna Singh et al. performed a DPP-IV enzyme assay of coumarin, and it displayed 54.83 nmol/mL of IC_50_ value [44]. The structures of coumarin derivatives reported as potential DPP-IV inhibitors by different researchers are illustrated in Figure 1.

The sulphonamide moiety (-SO_2_NH_2_) is a powerful pharmacophore in medicinal chemistry and drug discovery due to its broad range of biological activity [45]. In addition to creating hydrogen bonds with the residues of amino acids in the active pocket of biological targets, the sulfonyl functionality also joins with the structural core ring and constricts the side chains, enabling them to take on the particular conformations required for a snug fit. Omarigliptin is also a DPP-4 inhibitor with a sulfonamide portion, as depicted in Figure 1. Omarigliptin is distinct from other antidiabetic drugs since it has a long half-life, may be taken orally, and only needs to be dosed once every seven days [46].

To better comprehend the drug-receptor interaction, the current drug development industry often uses the molecular docking approach. Small therapeutic compounds’ binding affinities and orientations at the target site have been predicted using this method. Docking research has two primary goals: building precise models of structures and making reliable predictions of their activities. The development of a novel rational method to drug design based on macromolecular docking studies, in which drug structures are generated based on their fit to the 3D structures of a receptor site, has provided the most in-depth picture yet of the interaction between drugs and their targets [47]. Recent research on DPP-IV inhibitors has made it feasible to treat T2DM with minimal side effects. Therefore, in the present investigation, we aimed to discover and develop some coumarin-based sulphonamides as potential DPP-IV inhibitors in light of the fact that molecular hybridization of many bioactive pharmacophores frequently results in synergistic activity. The design approach for the derivatives is depicted in Figure 2. Each of the proposed derivatives was subjected to an in silico virtual screening, and those that passed all of the criteria and demonstrated higher binding affinity with the DPP-IV enzyme were then synthesized and tested in vitro.

## 2. Results and Discussion

### 2.1. Pre-ADME Analysis and Toxicity Profile

The process of developing novel medications takes a lot of time and resources because of how intricate it is. However, there is a significant attrition rate in medication development when investigators discard promising compounds because they either do not work as expected or present unacceptable hazards to humans. Potential pharmaceutical molecules are characterized using ADMET (absorption, distribution, metabolism, elimination, and toxicity) testing, which identifies both promising compounds and those with substantial limitations [48,49,50]. Each and every one of the proposed derivatives was put through an in silico ADMET analysis, and those found inferior were eliminated from the further screening process.

According to the principle put forward by Lipinski and Veber (Appendix A in the Supplementary Information), none of the molecules violated the Lipinski rule of five, but only one molecule violated Veber’s rule. The log *p* values of all of the compounds discovered ranged from 1.64 to 3.29, which indicates that they have excellent lipophilicity. Lipophilicity is an important property of the molecule that plays a role in determining how it functions inside the body [51]. It is defined by the log *p* value of the chemical, which evaluates the permeability of the medicine in the body in order for it to reach the target tissue [52,53]. All of the compounds had a molecular weight that was less than 500 Da, which suggests that they can move more easily across biological membranes. It was a fortunate coincidence that the Lipinski rule of 5 had not been broken by any of the compounds, including the native ligand [54,55]. It has been shown that the total polar surface area (TPSA), in addition to the number of rotatable bonds, is a stronger indicator of whether a molecule is orally active or not. Molecule 6f was found to be in violation of Veber’s rule, since it had a TPSA that was more than 140 Å^2^, or 142.96, which indicates that it has a low oral bioavailability.

The pharmacokinetics and drug-likeness features of each molecule were analyzed and estimated so that additional improvements might be made to the compounds (see Appendix A in the Supplementary Information). There was no evidence that any of the compounds, with the exception of the native ligand, could cross the blood-brain barrier (BBB). The values of all of the compounds’ log *Kp* (skin penetration, cm/s) and bioavailability were well within the range of permissible ranges. There are very few molecules that do not fit all three of Ghose, Egan, and Muegge’s conditions, or at least two of them (Appendix A). As more compounds are discovered to be inhibitors of cytochrome enzymes, thisraises concerns that they may also affect the metabolism of other medications. Due to its poor GI absorption, molecule 6e was not subjected to molecular docking experiments and was therefore excluded from the screening process.

In acute toxicity predictions (see Appendix A in Supplementary Information), molecules 6h, 6m–6q, and native ligand fall in toxicity class III (toxic if swallowed (50 < LD_50_ ≤ 300)). 6i, and 6l displayed toxicity class IV (harmful if swallowed (300 < LD_50_ ≤ 2000)). Molecules 6a–6g, 6k, 6r, and 6s showed toxicity class V (may be harmful if swallowed (2000 < LD_50_ ≤ 5000)). 6j and 6t displayed toxicity class VI (non-toxic (LD_50_ > 5000)) [56].

In the physicochemical radar images (Figure 3), all the molecules violated just one criterion, i.e., instauration (0.25 < Fraction Csp3 < 1) except molecules 6m, 6n, and native ligand. The BOILED-EGG model provides a quick, simple, easily repeatable, but statistically unparalleled robust strategy for forecasting the passive gastrointestinal absorption and brain access of tiny compounds relevant for drug discovery and development. The physical and chemical space of molecules with the greatest chance of being absorbed by the digestive system is represented by the white area, and the physical and chemical space of molecules with the greatest chance of penetrating to the brain is represented by the yellow area (the yolk) [57]. None of the designed molecules, including native ligands, displayed permeation through the blood-brain barrier, whereas all were predicted to be passively absorbed by the gastrointestinal tract. All the designed molecules were predicted not to be effluxed from the central nervous system by the P-glycoprotein, whereas, the native ligand predicted to be effluxed from the central nervous system by the P-glycoprotein (Figure 4). Based on the results of this virtual screening, it was determined that 6a–6d, 6g, 6i–6l, and 6r–6t possess drug-like characteristics; as a result, these compounds were investigated further using molecular docking.

### 2.2. Computational Analysis

The results of the pre-ADME and pre-tox predictions indicated that compounds 6e, 6f, 6h, and 6m to 6q were inferior and violated the most drug-likeness criteria. Molecular docking experiments, however, were not performed on these compounds. All of the docked derivatives’ binding affinities have been compared to the binding mode of the reference inhibitor [4-[(3*R*)-3-amino-4-(2,4,5-trifluorophenyl)butanoyl]piperazin-2-one] reported from the DPP-IV enzyme crystal structure (PDB ID: 5Y7H). The binding cavity of the enzyme was identified (grid box size (size_x = 27.9873862131 Å, size_y = 26.1213349882 Å, size_z = 33.0219279569 Å)) where the reference inhibitor was bound in the crystal structure, and the same cavity was selected to understand the binding affinity and binding interactions of designed molecules. There are many DPP-IV crystal structures available on the PDB site, but this one was chosen because it was the most recently reported. Compared to the reference inhibitor, several of the docked compounds showed more robust interactions and binding energy. Below, we explain and illustrate in Figure 5 and Figure 6, the docking interactions of the most effective compounds.

This time, the reference inhibitor was docked with the enzyme, and its results were used to judge the new docking results of the suggested compounds. The reference inhibitor had a binding free energy of −7.6 Kcal·mol^−1^ and formed an electrostatic salt bridge with Glu205 and Glu206. It has formed two conventional hydrogen bonds with Ser209 and Glu205. It displayed four conventional hydrogen bonds through fluorine with Arg125, Tyr666, and Asn710, and one carbon-hydrogen bond with Arg358. It also showed two halogen-fluorine bonds with Asn710 and His740. It has developed hydrophobic interactions (π-π stacked, π-π T-shaped, and amide-π stacked) with Tyr662, Tyr666, Ser630, and Tyr631. 6a displayed a −8 Kcal·mol^−1^ binding affinity and formed one conventional hydrogen bond with Arg356 and one carbon-hydrogen bond with Arg358. It also developed some hydrophobic interactions (π-π stacked, π-π T-shaped, and π-alkyl) with Phe357 and Arg356.

6b had a binding affinity of −8.5 Kcal·mol^−1^ and formed one conventional and one carbon-hydrogen bond with Arg125 and Arg358. It formed hydrophobic π-π stacked interactions with Phe357 and Tyr666. 6c developed only hydrophobic (π-sigma, π-π stacked, and π-alkyl) interactions with Tyr666, Phe357, Tyr631, and Tyr662 with −8.5 Kcal·mol^−1^ binding free energy. Despite failing to create a hydrogen bond with the enzyme, a stable complex was formed with good binding free energy. 6d had a docking score of −8.3 Kcal·mol^−1^ and formed one carbon-hydrogen bond with Arg358 and one π-sulfur bond with Tyr547. It has developed some hydrophobic π-π stacked, π-π T-shaped interactions with Phe357, Tyr662, and Tyr666. 6i developed three conventional hydrogen bonds with Val207, Arg356, and Arg358 with a binding affinity of −9.1 Kcal·mol^−1^. It showed one electrostatic π-cation bond with Arg358 as well as three hydrophobic π-π-stacked π-alkyl bonds with Phe357 and Arg356.

6j had a binding free energy of −9.3 kcal mol1 and displayed the same interactions as 6i. Both of these molecules showed the same binding mode in the active pocket of the target enzyme. With −8.3 Kcal·mol^−1^ binding free energy, 6r formed one conventional and one carbon-hydrogen bond with Arg356 and Arg358, respectively. Many hydrophobic π-π stacked, π-alkyl bonds have formed with Phe357, Arg356, Arg358, and Tyr666. 6s released −8.1 Kcal·mol^−1^ of binding free energy and formed one carbon-hydrogen bond with Arg358. It has developed many hydrophobic π-π stacked, π-π T-shaped, and π-alkyl bonds with Phe357 and Arg356. 6t formed one carbon-hydrogen bond with Arg358 and displayed a −8.7 Kcal·mol^−1^ binding affinity. It has also formed one π-sulfur bond with Tyr547. It has developed many hydrophobic π-π stacked, π-π T-shaped, and amide-π stacked with Phe357, Tyr662, Tyr666, Ser630, and Tyr631. Although the reference inhibitor and the developed compounds revealed slightly different binding behaviors due to the structural variations between them, it was intriguing to note that they both bound in the same binding pocket, as shown in Figure 5. The reference inhibitor and the docked compounds revealed one interaction in common with Arg358, however, the majority of the created molecules had the majority of the same interactions and shared the same amino acids. Each and every one of them developed stacking π-π interactions with Phe357. The findings presented above provide conclusive evidence that all of the ligands exhibited binding in the same cavity. From the above investigation, it was observed that, 6a, 6b, 6c, 6d, 6i, 6j, 6r, 6s, and 6t displayed more binding affinity than the reference inhibitor and were hence selected for wet lab synthesis, and the structures were confirmed by spectral analysis.

### 2.3. In Vitro Enzyme Assay

Table 1 shows the % inhibition of standard (Sitagliptin at 100 µM) and synthesized compounds (at 250 µM) along with IC_50_ (µM) values. Sitagliptin displayed a 0.018 µM IC_50_ value, which is the most potent. Also, all the tested compounds displayed significant inhibition of the DPP-IV enzyme, but 6i and 6j demonstrated 10.98 and 10.14 µM IC_50_ values, respectively, i.e., most potent among the synthesized compounds. It was observed that in computational analysis, 6i and 6j developed potent conventional hydrogen bonds with critical amino acids and also displayed an optimum binding affinity with the target. Fortunately, the results obtained here are completely consistent with those obtained from in silico screening.

The conclusion regarding the structural-activity relationship (SAR) can be effectively drawn from in silico screening and in vitro enzyme assays. It was observed in computational analysis that increasing carbon length beyond two carbon atoms decreases the activity (6l). A molecule with an unsubstituted phenyl ring (6a) increases the activity, while substitution at any position in the ring results in decreased activity. Substitution at the –Ar position with the benzimidazole group increases the activity (6i and 6j). Molecules with pyridyl substitution (6b–6d) increase binding affinity with the target enzyme and also display significant inhibition of DPP-IV. There is much scope to design more novel derivatives using the same nucleus by considering the above-discussed SAR. The SAR is illustrated in Figure 7.

## 3. Material and Methods

### 3.1. Pre-ADME Analysis, Toxicity Profile, and Computational Analysis

The Lipinski rule of five as well as the pharmacokinetic properties of the designed molecules were calculated with the help of the molinspiration and SwissADME servers [58,59]. ProTox-II is a web server that is freely accessible to the public and has been used in order to carry out an in silico toxicity prediction of proposed derivatives (http://tox.charite.de/protox_II, accessed on 20 October 2022) [56].

The molecular docking studies were performed using PyRx Virtual Screening Tool 0.8 (Source Forge Headquarters, 225 Broadway Suite 1600, San Diego, CA 92101) using the Autodock Vina 1.1.2 from Vina Wizard tool [60] and the Biovia Discovery studio was used to analyze the binding interactions between ligands and target enzyme [61]. The structures of ligands were drawn using ChemDraw Ultra 12.0 version (PerkinElmer, Inc., USA). The structures were imported in PyRx software 0.8 (Source Forge Headquarters, 225 Broadway Suite 1600, San Diego, CA 92101) with the help of an open bable toolbar and the energy minimization was executed by Universal Force Field (UFF) [62]. The RCSB Protein Data Bank was searched in order to obtain the crystal structure of the human DPP-IV in association with inhibitor-3 (PDB ID: 5Y7H) (https://www.rcsb.org/structure/5Y7H, accessed on 6 November 2022). The obtained enzyme structure was purified by deleting all the heteroatoms and water molecules with the help of Biovia Discovery studio. The purified structure was saved again in pdb file format for further use. The three-dimensional view of the DPP-IV enzyme with reference inhibitor in the active binding pocket is illustrated in Figure 8. The reference inhibitor present in the enzyme cavity was re-docked and its binding mode and binding energy were used to validate the docking results of screened compounds. Chain A was selected and the active cavity was identified using Biovia Discovery studio and the same cavity with the exact grid box (size_x = 27.9873862131 Å, size_y = 26.1213349882 Å, size_z = 33.0219279569 Å) was selected for docking with an exhaustiveness value of 8. After molecular docking was performed, the raw pdbqt files were used to study how the ligand and target interact [45,51,63,64,65,66].

### 3.2. Chemistry

Throughout this study, we used chemicals and reagents acquired from Lab Trading Laboratories in Aurangabad, Maharashtra, India. Spots were viewed using UV light and iodine vapors in a technique called thin-layer chromatography (TLC) to track the reaction’s progress. The open capillary method was used to determine the melting points (uncorrected) of the compounds. Using a JEOL 500 MHz spectrometer, DMSO was used as the solvent, and tetramethyl silane (TMS) was used as the internal standard, yielding ^1^H and ^13^C NMR spectra. Singlet (*s*), doublet (*d*), triplet (*t*), and multiplet (*m*) coupling frequencies; ^1^H-^1^H coupling constants in *Hertz*; chemical shifts stated in units or ppm. The synthesized compounds’ mass (*m/z*) spectra were collected using a Shimadzu LC-MS instrument and the spectrum is given in the supplementary information file. Using a Microlab IR Spectrophotometer from Agilent Technologies (5301 Stevens Creek Blvd., Santa Clara, CA 95051, USA), we were able to capture the infrared spectra of the produced molecules.

In a previously published paper, the step-by-step procedure for the synthesis of compounds 2, 3, 4, and 5 has already been explained [67]. The following section includes an explanation for the synthesis of 6a, 6b, 6c, 6d, 6i, 6j, 6r, 6s, and 6t:

In 20 mL ethanol and 5 mL DMF, a combination of compound 5 (3 mmol), suitable aryl chloride (3 mmol), and K_2_CO_3_ (3 mmol) were refluxed. TLC was used to monitor the reaction’s completion. The surplus solvent was dumped onto the ice and the pH was corrected to pH 7.0. (6 to 8). Vacuum filtration was used to collect the produced solid. Column chromatography was used to purify the crude solid, using EtAc/Pet. ether as the eluent (40:60). Physical characterization and recrystallization of the obtained products were carried out [68]. The step-by-step reaction route that followed in order to produce coumarin-based sulphonamide derivatives is shown in Figure 9.

In the next part, along with their IUPAC nomenclature, the spectral data and physical characterization of the compounds are given.

#### 3.2.1. 6a. [3-Chloro-2-oxo-2*H*-1-benzopyran-6-(*N*-phenyl)sulfonamide]

Off-white solid, yield: 61%, molecular formula: C_15_H_10_ClNO_4_S, melting point: 158–160 °C. Elemental analysis (cal.): C, 53.66; H, 3.00; Cl, 10.56; N, 4.17; O, 19.06; S, 9.55. FT-IR (neat, cm^−1^) ν_max_: 3480 (NH stretch), 3115 (NH bend w), 2956 (Ar stretch), 1765 (C=O stretch), 760 (C=O bend). ^1^H NMR (300 MHz, DMSO-d_6_, chemical shift (ppm)); δ 4.012 (s, NH of sulfonamide), 6.689, 6.712, 7.171, 7.281, 7.389, 7.478, 7.603, 7.732, 7.891 (m, *J* = 7.67, Ar-H). ^13^C NMR (400 MHz, DMSO-d_6_, chemical shift (ppm)): δ 116.92, 117.82, 118.23, 119.53, 120.22, 120.73, 121.87, 128.67, 129.80, 130.02, 138.92, 139.23, 142.64, 153.67, 159.02, 163.12. MS *m/z*: 336.18, 337.90 (m + 1), 340.23 (m + 2), 341.98 (m + 3).

#### 3.2.2. 6b. [3-Chloro-2-oxo-*N*-(pyridin-2-yl)-2*H*-chromene-6-sulfonamide]

Pale yellow, yield: 60%, molecular formula: C_14_H_9_ClN_2_O_4_S, melting point: 169–171 °C. Elemental analysis (cal.): C, 49.93; H, 2.69; Cl, 10.53; N, 8.32; O, 19.00; S, 9.52. FT-IR (neat, cm^−1^) ν_max_: 3440 (NH stretch), 3124 (NH bend w), 2928 (Ar stretch), 1710 (C=O stretch), 778 (C=O bend). ^1^H NMR (300 MHz, DMSO-d_6_, chemical shift (ppm)); δ 4.0 (s, NH of sulfonamide), 6.70 (s, NH of benzimidazole), 7.55, 7.70, 7.98, 8.07, 8.12, 8.30 (m, *J* = 7.51, Ar-H). ^13^C NMR (400 MHz, DMSO-d_6_, chemical shift (ppm)): δ 109.9, 117.9, 119.6, 120.8, 121.8, 122.5, 124.7, 136.4, 138.3, 140.7, 148.1, 152.9, 156.2, 156.5. MS *m/z*: 337.45, 338.76 (m + 1), 339.03 (m + 2).

#### 3.2.3. 6c. [3-Chloro-2-oxo-*N*-(pyridin-3-yl)-2*H*-chromene-6-sulfonamide]

Yellow, yield: 55%, molecular formula: C_14_H_9_ClN_2_O_4_S, melting point: 163–169 °C. Elemental analysis (cal.): C, 49.93; H, 2.69; Cl, 10.53; N, 8.32; O, 19.00; S, 9.52. FT-IR (neat, cm^−1^) ν_max_: 3445 (NH stretch), 3100 (NH bend w), 2916 (Ar-stretch), 1740 (C=O stretch), 698 (C=O bend). ^1^H NMR (300 MHz, DMSO-d_6_, chemical shift (ppm)); δ 4.0 (s, NH of sulfonamide), 5.089 (s, NH of benzimidazole), 7.27, 7.36, 7.70, 7.98, 8.04, 8.09, 8.12, 8.30 (m, *J* = 7.61, Ar-H). ^13^C NMR (400 MHz, DMSO-d_6_, chemical shift (ppm)):119.6, 120.8, 121.8, 122.5, 124.7, 136.4, 137.5, 138.8, 140.7, 156.2, 156.5. MS *m/z*: 336.23 (m + 1), 337.67 (m + 2), 338.43 (m + 3).

#### 3.2.4. 6d. [3-Chloro-2-oxo-*N*-(pyridin-4-yl)-2*H*-chromene-6-sulfonamide]

Yellow, yield: 67%, molecular formula: C_14_H_9_ClN_2_O_4_S, melting point: 168–169 °C. Elemental analysis (cal.): C, 49.93; H, 2.69; Cl, 10.53; N, 8.32; O, 19.00; S, 9.52. FT-IR (neat, cm^−1^) ν_max_: 3446 (NH stretch), 3110 (NH bend w), 2912 (Ar-stretch), 1710 (C=O stretch), 699 (C=O bend). ^1^H NMR (300 MHz, DMSO-d_6_, chemical shift (ppm)); δ 4.0 (s, NH of sulfonamide), 5.089 (s, NH of benzimidazole), 6.99, 7.70, 7.98, 8.12, 8.46 (m, *J* = 7.53, Ar-H). ^13^C NMR (400 MHz, DMSO-d_6_, chemical shift (ppm)):109.0, 119.6, 120.8, 121.8, 122.5, 124.7, 136.4, 140.7, 150.2, 156.2, 156.5. MS *m/z*: 337.75 (m + 1), 339.12 (m + 2), 340.15 (m + 3).

#### 3.2.5. 6i. [3-Chloro-2-oxo-2*H*-1-benzopyran-6-(*N*-benzimidazole)sulfonamide]

White crystalline solid, yield: 58%, molecular formula: C_16_H_10_ClN_3_O_4_S, melting point: 168–170 °C. Elemental analysis (cal.): C, 51.14; H, 2.68; Cl, 9.43; N, 11.18; O, 17.03; S, 8.53. FT-IR (neat, cm^−1^) ν_max_: 3468 (NH stretch), 3130 (NH bend w), 2945 (Ar stretch), 1742 (C=O stretch), 778 (C=O bend). ^1^H NMR (300 MHz, DMSO-d_6_, chemical shift (ppm)); δ 4.122 (s, NH of sulfonamide), 5.089 (s, NH of benzimidazole), 7.127, 7.234, 7.367, 7.452, 7.509, 7.639, 7.790, 7.821, 7.876, 7.920 (m, *J* = 7.48, Ar-H). ^13^C NMR (400 MHz, DMSO-d_6_, chemical shift (ppm)): δ 24.89, 115.72, 120.34, 121.67, 122.21, 123.45, 124.78, 125.89, 126.29, 128.45, 138.90, 139.87, 140.23, 141.56, 150.62, 153.76, 159.56. MS *m/z*: 375.19, 376.94 (m + 1), 377.67 (m + 2), 379.43 (m + 3).

#### 3.2.6. 6j. [3-Chloro-2-oxo-2*H*-1-benzopyran-6-(*N*-(5-methyl)benzimidazole)sulfonamide]

White crystalline solid, yield: 54%, molecular formula: C_17_H_12_ClN_3_O_4_S, melting point: 174–176 °C. Elemental analysis (cal.): C, 52.38; H, 3.10; Cl, 9.09; N, 10.78; O, 16.42; S, 8.23. FT-IR (neat, cm^−1^) ν_max_: 3462 (NH stretch), 3132 (NH bend w), 2942 (Ar stretch), 1734 (C=O stretch), 780 (C=O bend). ^1^H NMR (300 MHz, DMSO-d_6_, chemical shift (ppm)); δ 2.456 (s, methyl proton), 4.142 (s, NH of sulfonamide), 5.183 (s, NH of benzimidazole), 7.137, 7.244, 7.377, 7.552, 7.539, 7.669, 7.780, 7.841, 7.879, 7.972 (m, *J* = 7.56, Ar-H). ^13^C NMR (400 MHz, DMSO-d_6_, chemical shift (ppm)): δ 116.02, 120.44, 121.87, 122.29, 123.40, 124.98, 125.09, 125.67, 128.75, 139.92, 139.97, 140.12, 141.50, 149.66, 152.77, 153.51, 159.92, 160.39. MS *m/z*: 389.76, 390.96 (m + 1), 392.69 (m + 2), 394.44 (m + 3).

#### 3.2.7. 6r. [3-Chloro-2-oxo-*N*-(p-tolyl)-2*H*-chromene-6-sulfonamide]

Yellowish brown puffy solid, yield: 49%, molecular formula: C_16_H_12_ClNO_4_S, melting point: 115–120 °C. Elemental analysis (cal.): C, 54.94; H, 3.46; Cl, 10.14; N, 4.00; O, 18.30; S, 9.17. FT-IR (neat, cm^−1^) ν_max_: 3366 (NH stretch), 3240 (NH bend w), 2940 (Ar stretch), 1730 (C=O stretch), 678 (C=O bend). ^1^H NMR (300 MHz, DMSO-d_6_, chemical shift (ppm)); δ 2.34 (t, methyl proton), 4.0 (s, NH of sulfonamide), 5.183 (s, NH of benzimidazole), 6.98, 7.02, 7.70, 7.98, 8.12, 8.30 (m, *J* = 7.45, Ar-H). ^13^C NMR (400 MHz, DMSO-d_6_, chemical shift (ppm)): δ 21.3, 119.5, 119.6, 120.8, 121.8, 122.5, 129.8, 131.2, 134.7, 140.2, 156.5. MS *m/z*: 349.79, 350.67 (m + 1), 351.53 (m + 2).

#### 3.2.8. 6s. [3-Chloro-2-oxo-*N*-(o-tolyl)-2*H*-chromene-6-sulfonamide]

Yellowish brown solid, yield: 59%, molecular formula: C_16_H_12_ClNO_4_S, melting point: 117–125 °C. Elemental analysis (cal.): C, 54.94; H, 3.46; Cl, 10.14; N, 4.00; O, 18.30; S, 9.17. FT-IR (neat, cm^−1^) ν_max_: 3332 (NH stretch), 3245 (NH bend w), 2959 (Ar stretch), 1710 (C=O stretch), 678 (C=O bend). ^1^H NMR (300 MHz, DMSO-d_6_, chemical shift (ppm)); δ 2.12 (t, methyl proton), 4.0 (s, NH of sulfonamide), 5.183 (s, NH of benzimidazole), 66.51, 6.69, 7.01, 7.15, 7.70, 7.98, 8.30 (m, *J* = 7.57, Ar-H). ^13^C NMR (400 MHz, DMSO-d_6_, chemical shift (ppm)): δ 17.3, 119.6, 120.8, 121.8, 122.5, 123.7, 124.7, 126.5, 131.3, 136.1, 136.4, 140.7, 156.2. MS *m/z*: 350.79, 351.67 (m + 1), 352.53 (m + 2).

#### 3.2.9. 6t. [*N*-benzyl-3-chloro-2-oxo-2*H*-chromene-6-sulfonamide]

Off-white solid, yield: 65%, molecular formula: C_16_H_12_ClNO_4_S, melting point: 120–126 °C. Elemental analysis (cal.): C, 54.94; H, 3.46; Cl, 10.14; N, 4.00; O, 18.30; S, 9.17. FT-IR (neat, cm^−1^) ν_max_: 3390 (NH stretch), 3240 (NH bend w), 2950 (Ar stretch), 1730 (C=O stretch), 680 (C=O bend). ^1^H NMR (300 MHz, DMSO-d_6_, chemical shift (ppm)); δ 3.48 (t, methyl proton), 4.82 (s, NH of sulfonamide), 5.183 (s, NH of benzimidazole), 6.00, 6.12, 6.49, 7.13, 7.23, 7.26, 7.33 (m, *J* = 7.43, Ar-H). ^13^C NMR (400 MHz, DMSO-d_6_, chemical shift (ppm)): δ 46.2, 119.6, 120.8, 121.8, 122.5, 124.7, 126.7, 126.9, 128.5, 136.4, 140.7.156.2, 156.5. MS *m/z*: 351.67, 352.23 (m + 1), 353.78 (m + 2).

### 3.3. In Vitro Enzyme Assay

In vitro assays were used to test the potential inhibitory effects of the synthetic compounds on the DPP-IV enzyme. The DPP–IV inhibition experiment was carried out with the help of an enzyme assay kit (Cayman Chemical kit, item number: 700210). The detailed procedure to perform an in vitro enzyme assay is depicted in our recently published paper [67]. As a reference, sitagliptin was utilized as a DPP-IV inhibitor. For every compound examined, we calculated both its IC_50_ value in µM and its percentage of inhibition at 50 µM. In triplicate, the derivatives were examined, and the calculations were performed in GraphPad Prism (2365 Northside Dr. Suite 560, San Diego, CA 92108) [69].

## 4. Conclusions

DPP-IV inhibitors are the medications that have attracted widespread attention in the treatment of T2DM due to their proven effectiveness over the long term and their ability to better manage glucose levels. Coumarin derivatives have a broad variety of biological actions, including anticoagulant characteristics and cytotoxic effects on bacteria and other pathogens. These include anti-inflammatory properties, neuroprotection, antioxidant, and anti-hyperglycemic properties, as well as anti-adipogenic and neuroprotective properties. Recently, few researchers have reported some coumarin derivatives as potential DPP-IV inhibitors. In the present investigation, we aimed to discover and develop some coumarin-based sulphonamides as potential DPP-IV inhibitors in light of the fact that molecular hybridization of many bioactive pharmacophores frequently results in synergistic activity. All the tested compounds displayed significant inhibition of the DPP-IV enzyme, but 6i and 6j demonstrated 10.98 and 10.14 µM IC_50_ values, respectively, i.e., most potent amongst the synthesized compounds. Fortunately, the results obtained here are completely consistent with those obtained from in silico screening. There is huge scope to design more novel derivatives using the same nucleus by considering the SAR discussed in the article. In the near future, we are aiming to report the inhibitory potential of these derivatives on another enzyme that plays an important role in the regulation of blood glucose, which will give us better insight regarding the antidiabetic potential of developed molecules.

## Figures and Tables

**Figure 1 molecules-28-01004-f001:**
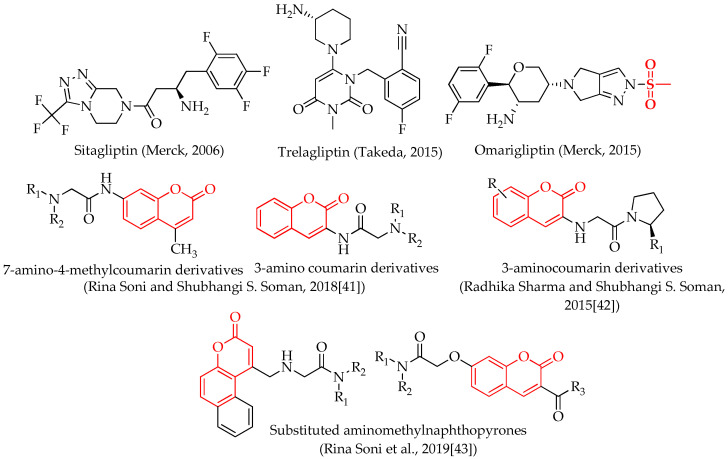
Structures of some approved drugs and coumarin derivatives reported by different researchers as DPP-IV inhibitors.

**Figure 2 molecules-28-01004-f002:**
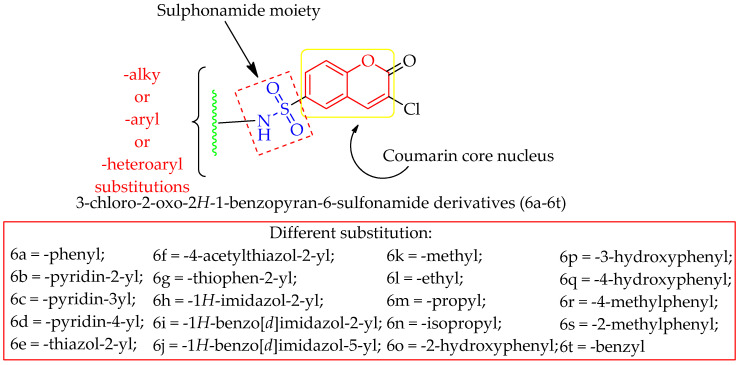
Designing an approach to develop some Coumarin-based sulphonamide derivatives as DPP-IV inhibitors.

**Figure 3 molecules-28-01004-f003:**
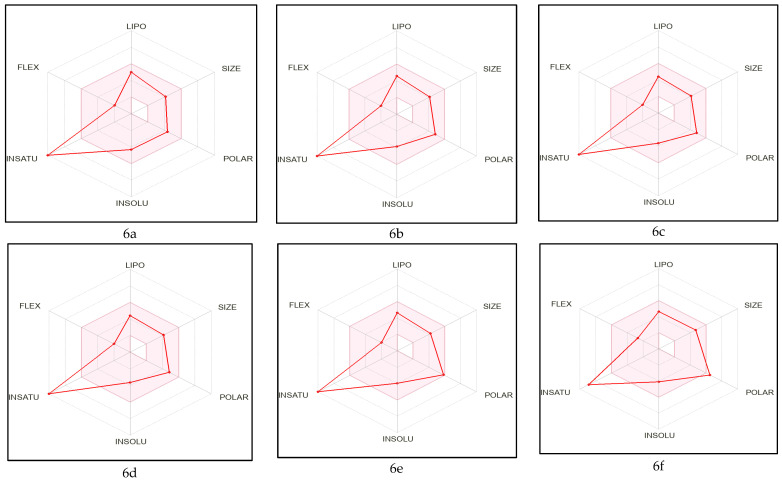
Physicochemical radar images of developed molecules where the colored zone is suitable for oral bioavailability obtained from the SwissADME online server.

**Figure 4 molecules-28-01004-f004:**
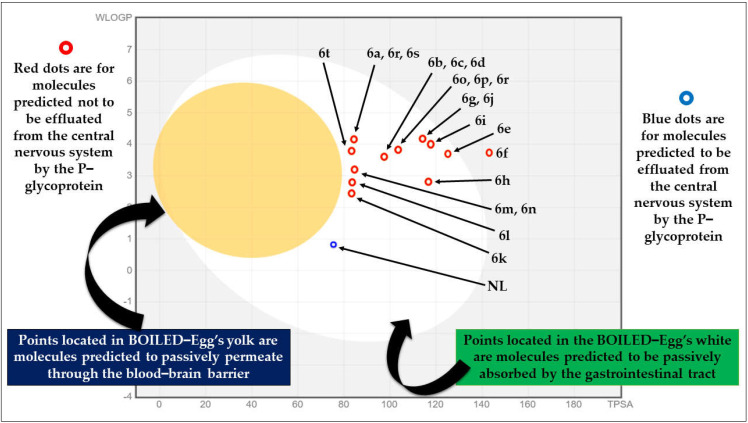
The BOILED-Egg plot of designed molecules obtained from the SwissADME online server, presented with modifications. (Note: BOILED-Egg plot of each molecule is given in Appendix A in the Supplementary Information).

**Figure 5 molecules-28-01004-f005:**
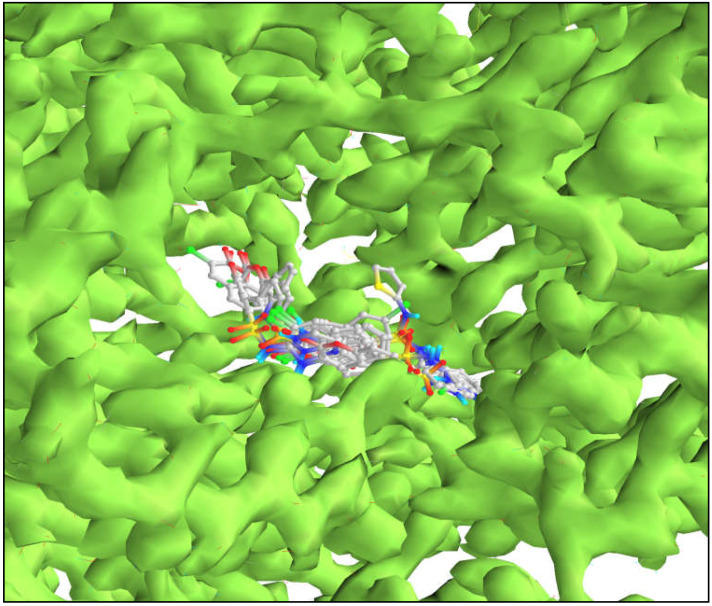
Molecular surface view of DPP-IV enzyme with ligands in active binding pocket showing similar conformational mode.

**Figure 6 molecules-28-01004-f006:**
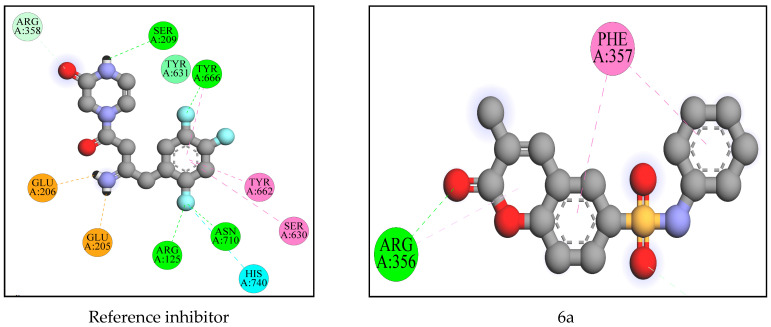
The molecular interactions of reference inhibitor and docked compounds which displayed less binding free energy than reference inhibitor.

**Figure 7 molecules-28-01004-f007:**
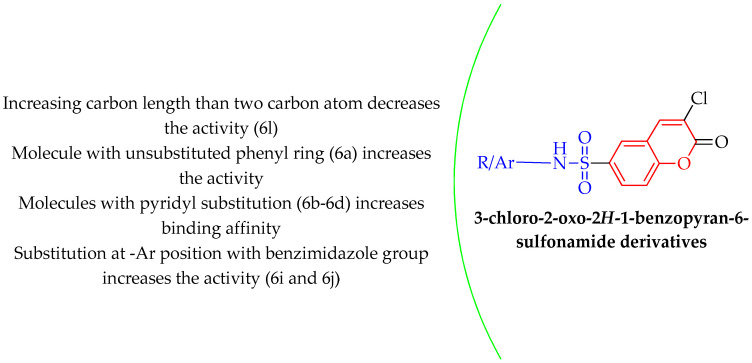
Structural-activity relationship of developed compounds.

**Figure 8 molecules-28-01004-f008:**
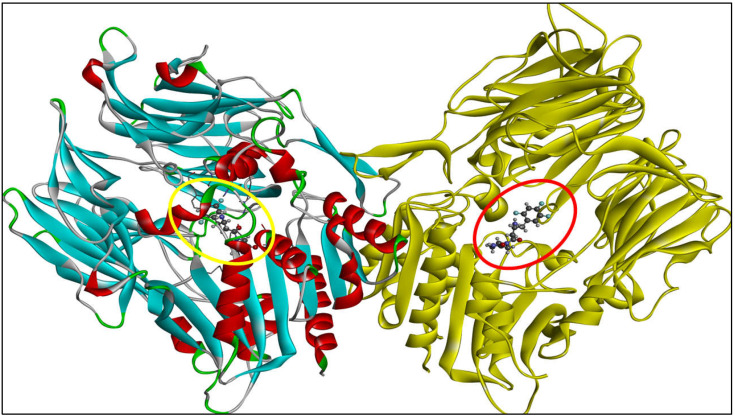
The 3D view of the target enzyme, Chain A displayed in red-sky blue color and Chain B exemplified in yellow color along with reference inhibitor present in the active binding pocket of both the chains denoted by yellow and red circle respectively.

**Figure 9 molecules-28-01004-f009:**
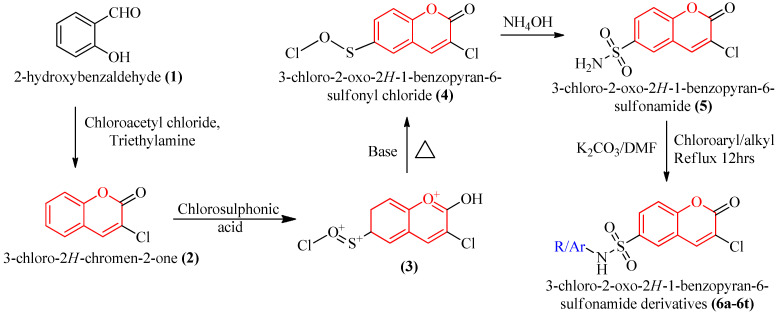
The detailed reaction pathway for the synthesis of coumarin-based sulphonamide derivatives.

**Table 1 molecules-28-01004-t001:** % inhibition of standard and synthesized compounds along with IC_50_ (µM) values (*n* = 3, mean ± SD).

Molecules Code	% Inhibition (at Conc.)	IC_50_ (µM)
Sitagliptin	102.6 ± 1.1 (100 µM)	0.018 ± 0.002
6a	93.52 ± 2.02 (250 µM)	14.52 ± 1.2
6b	91.15 ± 0.9 (250 µM)	15.72 ± 0.98
6c	90.34 ± 2.04 (250 µM)	16.28 ± 2.04
6d	92.40 ± 1.11 (250 µM)	15.19 ± 1.02
6i	95.77 ± 1.08 (250 µM)	10.98 ± 1
6j	96.59 ± 0.2 (250 µM)	10.14 ± 0.02
6r	95.44 ± 1.1 (250 µM)	12.92 ± 0.11
6s	89.33 ± 0.9 (250 µM)	18.29 ± 0.1
6t	88.29 ± 2.06 (250 µM)	17.73 ± 0.02

## Data Availability

Not applicable.

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
