# Peer review of "Coumarin-Based Sulfonamide Derivatives as Potential DPP-IV Inhibitors: Pre-ADME Analysis, Toxicity Profile, Computational Analysis, and In Vitro Enzyme Assay"

_molecules, 2023, doi:10.3390/molecules28031004_

Round 1
Reviewer 1 Report
The manuscript concerns a subject of interest which is the development of new antidiabetic drugs. However, it should be improved in several respects:
- page 2, line 65: insulin antagonists are not antidiabetic drugs;
- page 6, line 106 and Figure 2: the designing approach should be described more in detail; the same design approach is cited in the conclusions, therefore the conclusions should also be clarified in relation to the rational design;
- page 7, line 172: the name of the reference inhibitor present in PDB ID: 5Y7H should be included;
- page 8, Figure 3: in my opinion, the figure could be cropped and enlarged to better show the inhibitors, eliminating the part of the enzyme that does not interest;
- page 11, Table 4: the tested inhibitor concentrations that produced the reported inhibition % must be included in the table;
- page 11, line 245: compound 6l is not included in Table 4, therefore the reader cannot find any activity data to justify the described SAR;
- page 11, line 248 and Figure 5: "heteroaryl" should be replaced by "pyridyl";
- page 11, line 251 and Figure 5: why "predicted" SARs? In vitro enzyme assay was performed;
- English language should be revised.
Author Response
Dear Reviewer 1,
Thank you very much for your efforts to review our manuscript. The detailed response to the Reviewer 1 comments has been attached herewith

Reviewer 2 Report
I recommend "Reject" for this Manuscript due to the lack of rationality in molecular design and inadequate results of final products.
-They should give the meaning of T2DM in Introduction in first mentioned place.
-It is not obvious why methylsulfonyl grooup of Omarigliptin is red in Figure. This moiety is available lots of antidiabetic drugs.
-The -SO2NH2 group of Omarigliptin is in the ring. It is not true to use only one simple general group to develop similar structures. Is this group responsible for the activity of Omarigliptin? Authors at least should indicate this.
-I do not think the design of molecule is rational.
-If they use Swiss ADME, they should have used the radar diagrammatic representation and BOILED-Egg representation in manuscript.
-It is not clear why authors chose DPP-IV enzyme crystal structure (PDB ID: 5Y7H). Because there is not resemblance between compounds and ligand evogliptin in this enzyme.
-They should have showed the crucial residues in Figure 3. The 3D-chemical structures of these molecules are too smal compared to surface.
-For spectral analysis, they should put them in Supplementary data.
-What are the advantageous of these molecules to current therapy for further studies?
-
Author Response
Dear Reviewer-2,
Many thanks for your efforts in the evaluation of our manuscript. Please find our response to each comment. All the modifications in the revised manuscript have been indicated as yellow-colored texts. Please find the attached file for response details.

Reviewer 3 Report
Please see the attachment.

Author Response
Dear Reviewer,
Many thanks for your efforts in the evaluation of our manuscript. As per your comments, we have addressed all the issues to the point. Please find our response to each comment in the file attached herewith.

Round 2
Reviewer 3 Report
Although the authors made a remarkable effort to review the paper, there are still two important issues that must be addressed before publishing it in its current form.
1- 1- Mass spectra are not enough to validate the novelty of the synthesized compounds. 1H NMR and 13C NMR data must be added to the supplementary file.
2- The docking study should be modified to be persuasive to the reader. The synthesized compounds showed binding modes completely different from the co-crystallized ligand. In my opinion, the discussion is not completely convincing.
Author Response
Dear Reviewer,
Many thanks for your efforts in the evaluation of our manuscript. We strongly believe that your valuable observations, concerns, and suggestions will definitely improve the quality of our manuscript. As per your comments, we have addressed all the issues to the point. Please find our response to each comment. in the attached file. All the modifications in the revised manuscript have been indicated by highlighting with yellow color.
